# Preparation and Characterization of Screen-Printed Cu_2_S/PEDOT:PSS Hybrid Films for Flexible Thermoelectric Power Generator

**DOI:** 10.3390/nano12142430

**Published:** 2022-07-15

**Authors:** Junmei Zhao, Xiaolong Zhao, Rui Guo, Yaxin Zhao, Chenyu Yang, Liping Zhang, Dan Liu, Yifeng Ren

**Affiliations:** 1School of Electrical and Control Engineering, North University of China, Taiyuan 030051, China; zhaojunmei@nuc.edu.cn (J.Z.); zhaoxiaolong@nuc.edu.cn (X.Z.); yangwangcgsa@163.com (C.Y.); zhangliping@nuc.edu.cn (L.Z.); 2Key Laboratory of Instrumentation Science & Dynamic Measurement, Ministry of Education, North University of China, Taiyuan 030051, China; 18406589626@163.com (R.G.); zhaoyaxinnuc@163.com (Y.Z.)

**Keywords:** Cu_2_S, PEDOT:PSS, thermoelectric generator, screen-printing, fan-shaped

## Abstract

In recent years, flexible thermoelectric generators(f-TEG), which can generate electricity by environmental temperature difference and have low cost, have been widely concerned in self-powered energy devices for underground pipe network monitoring. This paper studied the Cu_2_S films by screen-printing. The effects of different proportions of p-type Cu_2_S/poly 3,4-ethylene dioxythiophene-polystyrene sulfonate (PEDOT:PSS) mixture on the thermoelectric properties of films were studied. The interfacial effect of the two materials, forming a superconducting layer on the surface of Cu_2_S, leads to the enhancement of film conductivity with the increase of PEDOT:PSS. In addition, the Seebeck coefficient decreases with the increase of PEDOT:PSS due to the excessive bandgap difference between the two materials. When the content ratio of Cu_2_S and PEDOT:PSS was 1:1.2, the prepared film had the optimal thermoelectric performance, with a maximum power factor (PF) of 20.60 μW·m^−1^·K^−1^. The conductivity reached 75% of the initial value after 1500 bending tests. In addition, a fully printed Te-free f-TEG with a fan-shaped structure by Cu_2_S and Ag_2_Se was constructed. When the temperature difference (ΔT) was 35 K, the output voltage of the f-TEG was 33.50 mV, and the maximum power was 163.20 nW. Thus, it is envisaged that large thermoelectric output can be obtained by building a multi-layer stacking f-TEG for continuous self-powered monitoring.

## 1. Introduction

With the accelerating urbanization process, the underground pipeline network is playing an irreplaceable role as the city’s blood vessel. To ensure the normal operation of the network, reducing personnel inspection tasks and saving costs, effectively monitoring the underground network is particularly important. Due to the many nodes and wide distribution of underground pipeline network, the traditional wireless monitoring system will be limited by the battery power, difficult to replace, high cost, and other outstanding problems. There is an urgent need for a maintenance-free, continuous power generation device that can use the environmental energy. As a device that can directly use the environmental temperature difference to generate electricity [1,2,3], the thermoelectric generator has no mechanical vibration structure [4], is maintenance-free [5,6], and can make full use of the cold and hot water pipes existing in the underground pipe network to generate electricity continuously, which will be the ideal power supply method for the long-term wireless monitoring system of the underground pipe network status. For this reason, the development and preparation of thermoelectric devices with high conversion efficiency is the key to their widespread use for wireless monitoring [7,8].

The conversion efficiency of a thermoelectric generator depends mainly on the figure of merit *ZT* (ZT=σT/k, where *S*, *σ*, T and *k* are Seebeck coefficient, electrical conductivity, absolute temperature, and thermal conductivity, respectively) of the thermoelectric materials [9]. To obtain a high thermoelectric figure of merit, the thermoelectric material should have a high power factor (PF=σ) and low thermal conductivity. Thus far, the majority of commercially available thermoelectric devices are based on Te-based thermoelectric materials, such as Bi_2_Te_3_ and PbTe. Researchers have also conducted numerous studies to improve thermoelectric properties [10,11,12,13,14]. For example, KUNIHISA et al. [15] reported Bi_0.4_Te_3.0_Sb_1.6_ films by spin-coating. The *ZT* of the films was 0.2 at 300 K. Pawan et al. [16] reported Bi_2_Te_3_ thin films by electrochemically deposited. The *S* of the film could be −20 μV·K^−1^. At the same time, the high toxicity and scarcity of Te element limit the widespread use of related thermoelectric materials. Therefore, the research and development of Te-free high-performance thermoelectric materials have become a direction of interest for most researchers [17,18,19,20,21]. Manisha et al. [22] reported an n-Type BiSe by pulsed electric current sintering. The high *ZT* was ~0.8 at 425 K. Ding et al. [23] prepared Ag_2_Se films by vacuum-assisted filtration. The *PF* of the film was 987.4 ± 104.1 μW·m^−1^·K^−2^ at 300 K. Among them, Cu_2_S as a P-type semiconductor with low thermal conductivity, high Seebeck coefficient, and excellent *ZT* has received much attention. Mulla et al. [24] prepared the Cu_2_S by copper ion doping. The *S* was 415 μV·K^−1^, given a high *PF* of about 400 μW·m^−1·^K^−2^. Cui et al. [25] reported a high thermoelectric figure of merit *ZT* of 1.2 could be achieved in hole-doped Cu_2_S crystal along the b-axis direction at 500 K. Instead, research on Cu_2_S has focused on the preparation of its high-performance bulk thermoelectric material, with the high cost and material waste in the preparation of the corresponding thermoelectric device [26,27], and the final device is difficult to effectively fit the surface of the pipe. For this reason, the film is one of the most effective ways to achieve rapid and low-cost preparation of its thermoelectric devices, especially flexible film [28,29,30]. The methods of vacuum-assisted filtration, magnetron sputtering, and electrochemical deposition have been used to prepare Cu_2_S flexible films [29,31,32,33]. For example, Liu et al. [29] prepared Cu_2_S films by vacuum-assisted filtration. The maximum *PF* of composite films was 56.15 μW·m^−1^·K^−2^ when Cu_2_S content was 10 wt% at 393 K. Liang et al. [34] prepared Cu_2_S by Ti4+ doping. The peak *ZT* value of 0.54 at 673 K. As a fast, very low-cost, graphic, and large-area method for screen printing to prepare thin films, there are few reports on Cu_2_S flexible thermoelectric films, although it has been widely used to prepare other thermoelectric films [35,36,37,38,39]. Therefore, the preparation of high-performance Cu_2_S thermoelectric films and graphic thermoelectric device structure design by screen printing is essential for developing Te-free thermoelectric devices and self-powered wireless monitoring by making full use of waste heat from underground pipe networks.

In this paper, Cu_2_S flexible films were prepared by screen printing. The effects of the content ratio of Cu_2_S and PEDOT:PSS on the structure, morphology, and thermoelectric properties of the films were systematically investigated. In addition, a fully printed Te-free flexible thermoelectric device with a fan-shaped structure was constructed. Its output performance was investigated in detail by combining the n-type Ag_2_Se thermoelectric film with superior performance prepared by our group’s previous work [35]. This TEG is expected to obtain high electrical output capability after being prepared by multi-layer stacking, which can provide an effective solution for continuous underground pipe network self-supply wireless monitoring.

## 2. Experimental Procedures

### 2.1. Material

The materials used in this experiment, Cu_2_S were purchased from Runyou Chemical Co., Ltd. (Nanjing, China), PEDOT:PSS was purchased from Shanghai Ouyi Organic Optoelectronic Materials (Shanghai, China), and anhydrous ethanol was purchased from Tianjin Kaitong Chemical Reagent Co. (Tianjin, China). All reagents were used directly without purification.

### 2.2. Preparation of Flexible Thermoelectric Thin Film

All containers and tools had been ultrasonic cleaning before the experiment. Firstly, 0.5 g Cu_2_S and different quantities of PEDOT:PSS were weighed and mixed thoroughly to prepare the printing paste. Polyimide (PI) was cleaned by ultrasonic and anhydrous ethanol to remove impurities from its surface. Secondly, the above printing paste was screen printed on the PI substrate with 200 mesh and cured at a constant temperature for 10 min. The printing and curing process was repeated three times until the printing paste was exhausted. By adjusting the content of PEDOT:PSS, the content ratio of Cu_2_S and PEDOT:PSS was determined to be 1:1.1, 1:1.2, 1:1.3, and 1:1.4, and they were named P1.1, P1.2, P1.3, and P1.4. After corresponding experiments, the best performance of the prepared films was obtained when the ratio of Cu_2_S and PEDOT:PSS was 1:1.2. Finally, the printing paste, prepared by a 30:1 ratio of Ag_2_Se to PVP, was printed on the P1.2 film to complete the f-TEG. The f-TEG consisted of 10 strips measuring 30 mm × 5 mm, arranged in a semicircular array on the PI. The f-TEG thermocouples were connected by conductive silver adhesive to reduce additional resistance. The manufacturing process using low-cost screen printing of thermoelectric films is shown in Figure 1.

### 2.3. Measurements and Characterizations of Cu_2_S/PEDOT:PSS Film and f-TEG

The physical-phase composition of Cu_2_S was measured using X-ray diffraction (XRD, DX-2700, Dandong, China). The composite films’ surface morphology and thickness were observed using a field emission scanning electron microscope (FESEM, SUPRA-55, Zeiss, Jena, Germany). Meanwhile, the X-ray energy spectrometer was analyzed. The Seebeck coefficient (*S*) and electrical resistivity (*σ*) were measured in a helium atmosphere (Linseis, LSR-3, Selb, Germany), with *S* and *σ* error measured by approximately ±5%. Carrier concentration (*n*) and mobility (*µ*) were determined by the Van der Pauw method under the nitrogen atmosphere (Linseis, HCS, Selb, Germany).

A test circuit with f-TEG as the power supply was established to test its output performance. The thermal side of the f-TEG obtained a variable high temperature through the heating table, and the cold side maintained a stable temperature through the water circulation cooling device. The output voltage could be measured by changing the heating temperature. The temperature was measured using a contactless infrared thermometer and an infrared imager.

## 3. Results and Discussion

The X-ray diffraction of Cu_2_S powder, films, and PEDOT:PSS is shown in Figure 2. The diffraction spectra of both films and powders have five distinct characteristic diffraction peaks with corresponding crystal planes (101), (103), (104), (113), and (200). The results are consistent with Cu_2_S in the standard spectrum. The results show that diffraction peaks of the films can be marked as Cu_2_S (PDF # 29-0578). Some other characteristic peaks are consistent with PDF # 06-0464 (CuS), which indicates a small amount of CuS impurities in the powder. In addition, the characteristic peaks of hybrid films do not correspond to Cu_2_S powder, and they correspond to the diffraction peaks of PEDOT:PSS films.

The SEM images of the P1.1, P1.2, P1.3, and P1.4 films are shown in Figure 3a–d. It could be seen that with the increase of PEDOT:PSS content, the interparticle spacing of Cu_2_S becomes large. The increase of a small amount of PEDOT may result in improved electrical properties of thin films. The thickness of the screen-printing film is shown in Appendix A. The measured thickness was 80–90 μm, indicating the uniformity of screen-printed hybrid films.

Table 1 showed the elemental ratios on the surface of P1.2 films. The result showed that the percentage of Cu and S was about 2:1, which conformed to the stoichiometric ratio. In addition, the P1.2 film element mapping results were shown in Appendix A, which indicated that Cu_2_S was uniformly distributed in the composite film. The results showed that the screen-printing method prepared a homogeneous composite film.

As shown in Figure 4, the *S*, *σ*, and *PF* of P1.1, P1.2, P1.3, and P1.4 films were tested, and their thermoelectric properties at 300–400 K were studied. In Figure 4a, the positive *S* indicated that Cu_2_S was a p-type TEG material. The *S* of these films showed similar trends, indicating that the addition of PEDOT:PSS did not affect the intrinsic properties of Cu_2_S. With the increase of PEDOT:PSS content, the *S* of the film gradually decreased, and with the temperature rise, the S of all films steadily increased. In Figure 4b, the *σ* of these films maintained a downward trend with increasing temperature. For example, the *σ* of P1.2 film decreased from 220.23 S·cm^−1^ at 300 K to 192.17 S·cm^−1^ at 400 K. With PEDOT:PSS increased, the *σ* of these films increased accordingly. For the composite film of Cu_2_S and PEDOT:PSS, the gap difference between Cu_2_S and PEDOT:PSS is large, and the interface barrier is large. Therefore, the S decreases with the increase of PEDOT:PSS content. The interaction between Cu_2_S and PEDOT forms a highly conductive PEDOT interface layer on the surface of Cu_2_S. The conductivity of this interface layer is much higher than that of PEDOT film and Cu_2_S, so the conductivity of the composite film increases. Figure 4c showed the *PF* of Cu_2_S/PEDOT:PSS composite films with different ratios varied with temperature. Under the *S* and *σ* combined, the *PF* of films with different ratios increased with increasing temperature. The film showed excellent thermoelectric value when the content ratio of Cu_2_S and PEDOT:PSS was 1:1.2. The P1.2 film had the most *PF* of 20.60 μW·m^−1^·K^−1^ at 400 K.

The Hall effect of the three thin films was tested using the Vanderbilt method to account for the *S* and *σ* with temperature. Since the variation patterns of *σ* and *S* were the same for all composite films in Figure 4, the variation pattern was explained here with the P1.2 composite film as the object of study. In Figure 5, the carrier concentration decreased from 4.38 × 10^22^ cm^−3^ to 2.17 × 10^22^ cm^−3^, and the mobility increased from 3.14 × 10^−2^ cm^−2^V^−1^s^−1^ to 5.52 × 10^−2^ cm^−2^V^−1^s^−1^ when the temperature increased from 300 K to 400 K. With the increase of temperature, Cu_2_S will undergo a structural phase transition process, that is, from non-cubic phase to cubic phase. In the high-temperature Cu_2_S cubic phase, Cu is not arranged in an orderly manner, which becomes a fast ion diffusion in the lattice, and the *S* sublattice provides a good transport channel. With the increase of temperature, the carrier velocity increases, the scattering effect decreases, so the mobility increases. PEDOT:PSS has metal characteristics, so the carrier concentration of the composite film decreases with the increase in temperature. The σ can be calculated according to the formula [40]:(1)σ=neμ
where *n* is the carrier concentration, *μ* is the mobility, and *e* is the electron mass. Therefore, the main reason for the decrease in *σ* with the temperature increase was that the increase in *μ* was significantly smaller than the *n* decrease. The *S* can be obtained according to the formula [40]:(2)S=8π2kB3eh2m*T(π3n)23
where *k_B_* is Boltzmann constant, *h* is Planck constant, and *m** is effective carrier mass. The decrease of *S* with the increase of PEDOT:PSS content was caused by the increase of *n* in the material. As the temperature increased, the reduction of *n* led to the rise of the *S*.

The f-TEG should be very flexible to tightly wrap the pipe to obtain more thermal energy. Therefore, to further verify the flexibility of the mixed film, this paper tested the flexibility of all proportions of the composite films. As shown in Appendix A, the dried P1.1 film no longer adhered to the PI substrate and had very poor flexibility. The flexibility test result was shown in Figure 6. In the experiment, the film was wound around a rod with a diameter of 8 mm, and the film was repeatedly bent. The internal resistance of each film increased with the increase of bending times. The results showed that after 1500 bends, the *σ* of the P1.4 film decreased to 84 % of the initial state, P1.3 to 80 %, and P1.2 to 75 %, meaning that films with higher PEDOT:PSS content had the best flexibility, so films with PEDOT:PSS had good bond flexibility.

After f-TEG was composed, the feasibility of f-TEG as a self-powered device was verified through practical tests. Considering the shape of the underground pipe, f-TEG is designed as a fan structure to fit the pipe wall better to obtain a higher output capacity by creating a larger temperature difference. To test the feasibility of the structure, COMSOL 5.6 software was used for the simulation experiment. The result was shown in Figure 7. The simulation experiments of the f-TEG were carried out in COMSOL software, and the corresponding simulation results were obtained. The output voltage of the f-TEG was 34.0 mV at the ΔT of 20 K and 60.0 mV at the ΔT of 35 K.

Based on the above test of Cu_2_S film, the f-TEG selected P1.2 as the p-type thermocouple. Our research group previously conducted the corresponding research on n-type thermocouples [35]. The group prepared the Ag_2_Se/PVP films. When the content ratio of Ag_2_Se and PVP was 30:1, the prepared film had the best thermoelectric properties with a maximum *PF* of 4.3 µW·m^−1^·K^−2^. So Ag_2_Se/PVP thermoelectric film was a reliable n-type thermocouple.

The real test was conducted in Figure 8a. The f-TEG’s output voltage and output power under a specific temperature difference could be measured by changing the temperature difference between the two ends of the device. Figure 8b was the infrared image of the f-TEG in the actual test. In the experiment, the inside of the device was the hot end, and the outside was the cold end—the schematic diagram of the test circuit as shown in Figure 8c,e. Test results for f-TEG were shown in Figure 8d,f. When the ΔT was 20 K and 35 K, the output voltages of the f-TEG were 19.3 mV and 33.5 mV. The actual test results were about 1.7 times smaller than the simulation results. This result may be caused by heat loss and contact resistance of conductive silver glue and wire. The maximum output powers of the f-TEG were 50.32 nW and 163.20 nW. Voltage *U_l_* could be calculated according to the formula [40]:(3)$$U_{l}=(1-\frac{R_{in}}{R_{in}+R_{l}})\times U_{0}$$
where *U*_0_ is the open-circuit voltage of the flexible thermoelectric device, and *R_in_* is its internal resistance. The formula showed that the output voltage increased with the increasing load resistance at a certain temperature difference. Eventually, it was infinitely close to the open-circuit voltage. When the load resistance was equal to the device’s internal resistance of 1750 Ω, the device’s output was 162.48 nW. The power density (*P_d_*) of the flexible thermoelectric device at the ΔT of 35 K was 0.0016 W·m^−2^, which could be calculated by the formula [40]:(4)$$P_{d}=\frac{P_{\max}}{N\times S}$$
where *P_max_* is the maximum output power, *N* is the number of the f-TEG bands, and S is the cross-sectional area. The results have shown that the flexible thermoelectric films prepared by screen-printing had excellent application potential in self-powered energy. For increasing output capability, the f-TEG can be prepared by multi-layer stacking; the method is easy to process. Figure 8g shows that f-TEG, prepared by the stacking method, will be the ideal self-power supply method for the long-term wireless monitoring system of the underground pipe network status.

## 4. Conclusions

In summary, the flexible thermoelectric films with different content ratios of Cu_2_S and PEDOT:PSS were prepared by screen printing. With the increase in the proportion of PEDOT:PSS, the *S* of the film decreased. The P1.2 film had the best thermoelectric properties, with a maximum PF of 20.60 μW·m^−1^·K^−1^. After 1500 times bending, the conductivity of the film is 75%, which indicates that the film has good toughness. In addition, the f-TEG prepared by Cu_2_S and Ag_2_Se was studied. When the ΔT was 35 K, the output voltage of the f-TEG was 33.50 mV, and the maximum output power was 163.20 nW. The f-TEG is expected to obtain high electrical output capability after being prepared by multi-layer stacking, which can provide an effective solution for continuous underground pipe network self-supply wireless monitoring.

## Figures and Tables

**Figure 1 nanomaterials-12-02430-f001:**
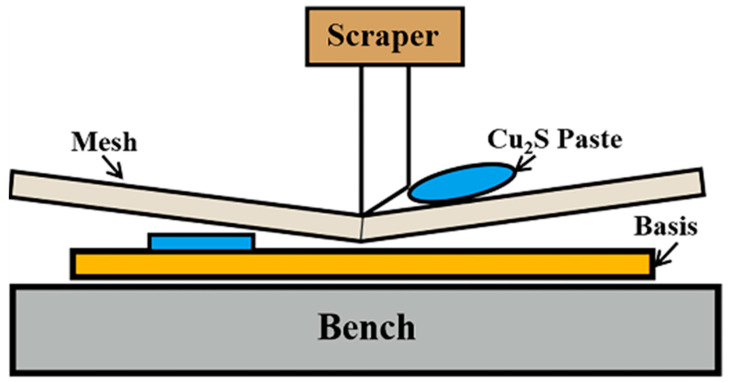
Screen printing diagram of Cu_2_S film.

**Figure 2 nanomaterials-12-02430-f002:**
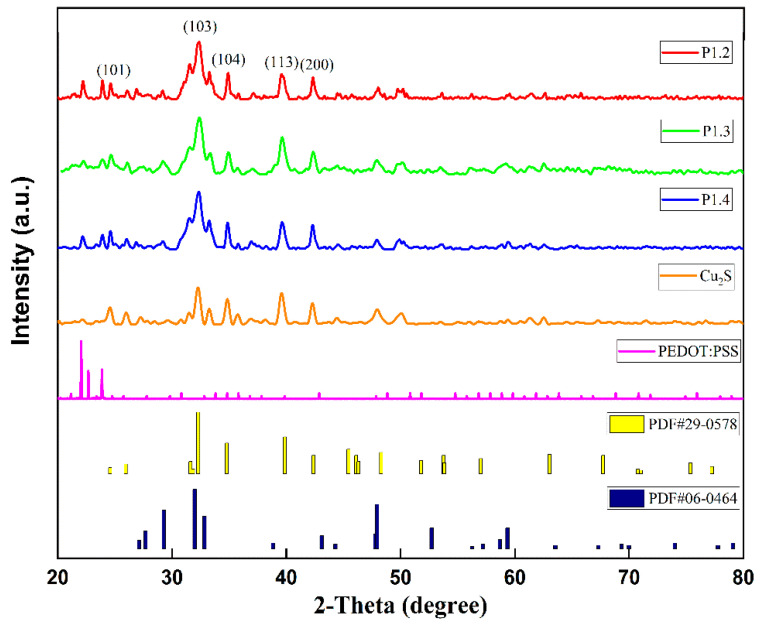
XRD patterns of powder and films.

**Figure 3 nanomaterials-12-02430-f003:**
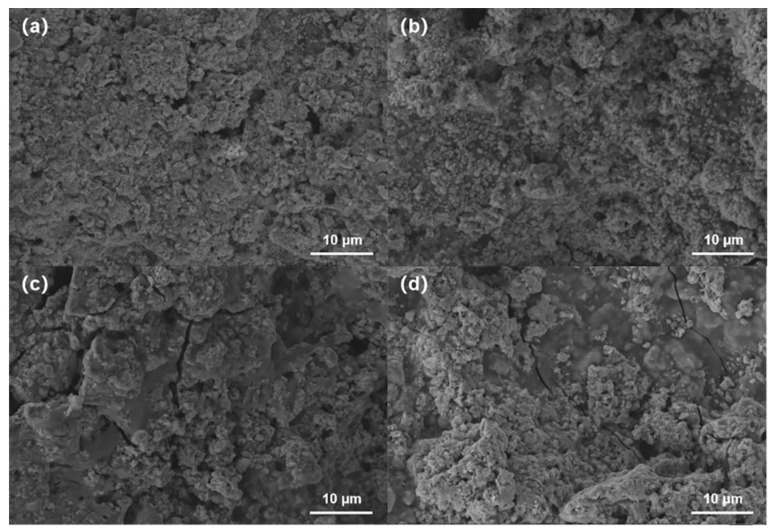
The SEM of (**a**) P1.1 film, (**b**) P1.2 film, (**c**) P1.3 film, (**d**) P1.4 film.

**Figure 4 nanomaterials-12-02430-f004:**
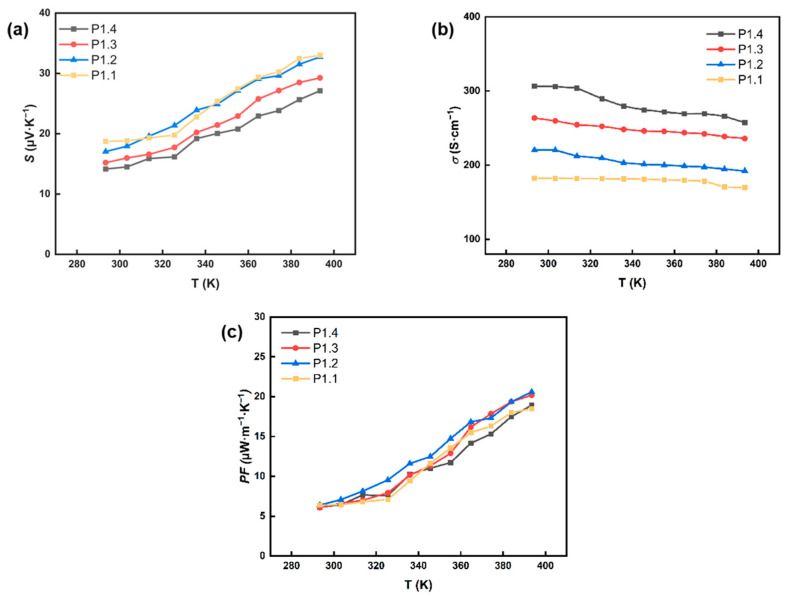
Relationship between temperature difference and (**a**) Seebeck coefficient (*S*), (**b**) electrical conductivity (*σ*), (**c**) Power factor (*PF*) of P1.1, P1.2, P1.3, and P1.4 films.

**Figure 5 nanomaterials-12-02430-f005:**
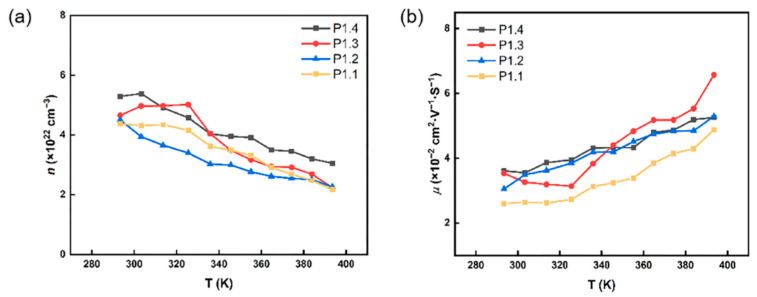
Temperature dependence of (**a**) carrier concentration (*n*) and (**b**) mobility (*μ*) of P1.1, P1.2, P1.3, and P1.4 films.

**Figure 6 nanomaterials-12-02430-f006:**
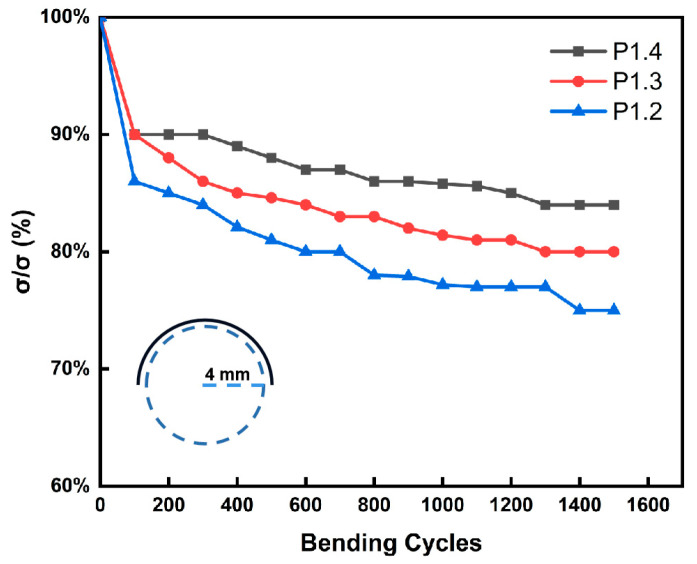
The flexibility of P 1.2, P1.3, and P1.4 films.

**Figure 7 nanomaterials-12-02430-f007:**
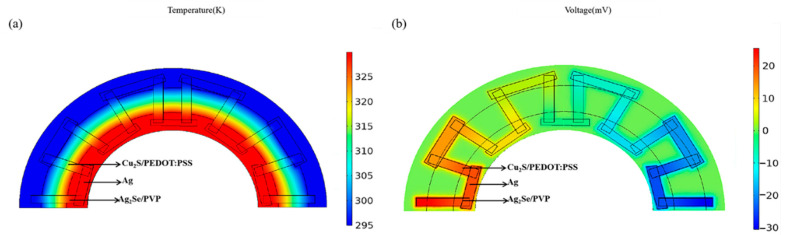
Simulation of f-TEG (**a**) temperature distribution and (**b**) output voltage.

**Figure 8 nanomaterials-12-02430-f008:**
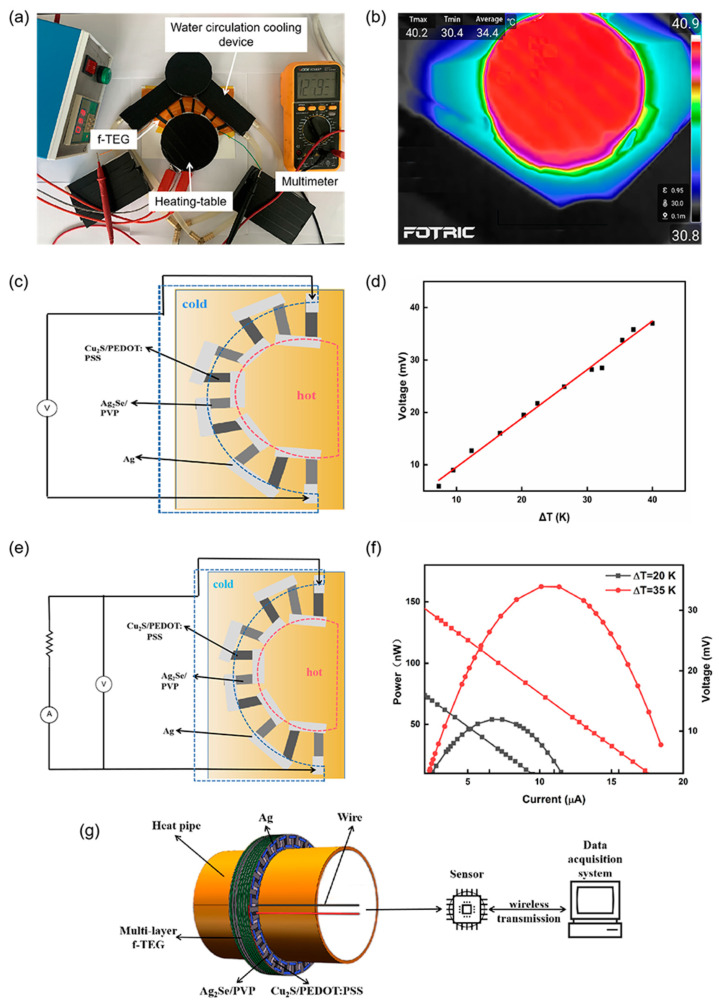
(**a**) Actual test diagram, (**b**) Infrared imager picture, (**c**) Schematic of output voltage test circuit, (**d**) Output voltage variation with temperature, (**e**) Schematic of output power test, (**f**) Schematic of voltage, current, and power relationship when ΔT is 20 K and 35 K, (**g**) Self-powered underground pipe network by f-TEG.

**Table 1 nanomaterials-12-02430-t001:** Elemental analysis results of the P1.2 film.

Element (%)	Weight (%)	Atom (%)
Cu	65.3	35.21
S	15.51	16.57
C	10.03	28.6
O	9.16	19.62
Total	100	100

## Data Availability

The raw/processed data required to reproduce these findings cannot be shared at this time due to legal or ethical reasons.

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
