# Peer review of "Preparation and Characterization of Screen-Printed Cu2S/PEDOT:PSS Hybrid Films for Flexible Thermoelectric Power Generator"

_nanomaterials, 2022, doi:10.3390/nano12142430_

Round 1

Reviewer 1 Report

The authors present a flexible thermoelectric generator based on Cu_2S and PEDOT:PSS for the p-leg and Ag_2S and PVP for the n-leg. They propose an application in which a multilayer version of the proposed generator could be used to power sensors monitoring underground pipes carrying hot fluids.

The concept of a flexible thermoelectric generator with reasonable efficiency is in principle interesting, but it is not clear why it should be convenient for the particular application proposed by the authors, since its flexibility is of no advantage here and also the reduced thickness is not of interest, since the idea is of stacking many layers on top of each other. The authors do not provide evidence that stacking many thin generators would provide an advantage with respect to a bulk generator with the same thickness.

The reason why this solution is appropriate for the particular application considered should be more convincingly argued.

Furthermore, from Fig. 8(a) it is not clear how good of a thermal contact exists between te cold sources and the cold sides of the thermoelectric generator: there seems to be limited overlap and in some case no overlap.  Thermal contact might take place via the substrate, but if this were a good thermal conductor, it would thermally short circuit the TEG.

Fig 8(b) does not seem to be consistent with what one would expect, i.e. the presence of a thermal gradient between the hot source (in the middle) and the cold sources (on the outside). Rather, there seems to be a peak temperature at an intermediate position. If this figure is right, it should be explained in detail.

In Figs. 8(c) and 8(e) the cold source appears to contact only two junctions and the same appears to happen for the hot source.

In addition, in the abstract the formation of a "superconducting layer" is mentioned, which is certainly impossible at the temperatures of operation of the mentioned devices.

Overall, the English should be improved, because there are many awkward expressions.

Author Response

Thanks for your review, please see the attachment.

Reviewer 2 Report

The paper reports a study of screen printed Cu2S/PEDOT:PSS films for flexible thermoelectric generators. The proposed materials and the proposed device appear interesting for applications. The film characterization and the device performances are well discussed. The conclusions are well supported by data. Moderate English corrections are necessary, thus I suggest to correct them. In conclusion, I suggest the paper to publication after moderate english revision. 
